# EVA: Achieving Discriminative and Semantically Faithful Multi-Scale EEG-Vision Alignment

## Abstract

Decoding semantic information from electroencephalography (EEG) signals elicited by diverse visual stimuli remains a critical challenge in brain-computer interfaces and cognitive neuroscience. Existing approaches typically align EEG with single-modality visual stimuli but struggle to generalize across multiple modalities and temporal scales. We propose EVA (EEG-Vision Alignment), the first framework that unifies multi-scale EEG alignment with heterogeneous visual stimuli, including rapid image presentations, continuous video sequences, and 3D object rotations, within a single contrastive learning-based architecture. EVA's Universal EEG Encoder features two key innovations: (1) a Frequency-Aware Dynamic Encoding (FADE) module that transforms EEG signals into the frequency domain via real-valued fast Fourier transform, enabling compact, adaptive representations through adjustable band-pass filtering; and (2) an Adaptive Channel Clustering (ACC) module that dynamically updates channel groupings using cross-attention and gradient-based optimization, capturing inter-channel synergies while mitigating noise. By optimizing EEG features to achieve both discriminative power for robust classification and semantic fidelity for high-quality reconstruction from brain signals, our framework achieves state-of-the-art performance across diverse tasks, including image retrieval, video classification, and 3D object recognition, on multiple datasets. Notably, our zero-shot reconstruction of 200 object categories from the THINGS-EEG dataset, using only aligned EEG features without textual or low-level cues, surpasses prior state-of-the-art by a significant margin. These results underscore EVA's capability to extract robust, generalizable representations from EEG signals, demonstrating the superiority of our unified framework. Code will be released upon publication.

## 1 Introduction

Understanding the neural mechanisms that underlie human visual cognition represents one of the most profound challenges in neuroscience and artificial intelligence (Van Essen et al., 1992; DiCarlo & Cox, 2007; Tsao et al., 2006). During visual processing, distinct patterns of electrical activity arise across the brain (Hebart et al., 2023), which can be measured non-invasively through electroencephalography (EEG) (Gifford et al., 2022; Liu et al., 2024b; Guo et al., 2024). These neural signatures contain rich semantic information about observed stimuli, yet decoding this information presents significant challenges due to EEG's high dimensionality and poor signal-to-noise ratio. Recent advances in neural recording technologies and the collection of relevant datasets have created opportunities to extract meaningful visual semantics from brain signals, with potential applications spanning assistive technologies and novel human-computer interaction paradigms (Benchetrit et al., 2023; Chen et al., 2024).

Recent advances in visual representation learning, particularly through contrastive learning approaches (Radford et al., 2021a; Zhai et al., 2023) and vision-language models (Li et al., 2023; Jia et al., 2021; Zhai et al., 2022), have demonstrated impressive zero-shot capabilities across diverse visual tasks. These powerful models offer a promising avenue for brain decoding: aligning neural signals with their semantic spaces could potentially unlock more effective neural decoding. While this approach has shown success in fMRI studies (Scotti et al., 2024; Gong et al., 2025), in the EEG

domain, this direction remains largely unexplored, with only a handful of studies attempting such alignment (Song et al., 2024; Li et al., 2024). These pioneering works face significant limitations: most employ simplistic encoders that fail to capture complex EEG dynamics, ignore multi-channel relationships, and typically target specific tasks or datasets. Current approaches lack the flexibility to handle diverse visual stimuli across varying temporal scales, and a unified framework capable of aligning EEG signals with different visual modalities (images, videos, 3D objects) remains elusive.

EEG data features high temporal resolution but low signal-to-noise ratio, with signals spanning multiple frequency bands that carry different cognitive information. Traditional time-domain encoding approaches (Zhang et al., 2023b; Altaheri et al., 2022; Zhang et al., 2022) often struggle to capture frequency-specific information in EEG signals, particularly those elicited by rapid visual stimuli. Additionally, existing approaches to handling the multi-channel nature of EEG are limited (Shi et al., 2023): Channel Dependent methods indiscriminately combine all channels, potentially causing over-smoothing (Song et al., 2022); Channel Independent methods process each channel separately, ignoring inter-channel relationships (Nie et al., 2022); Prior knowledge-based methods and hard clustering approaches remain fixed during training, unable to adapt to evolving data distributions and task requirements (Yi et al., 2023).

To address these challenges, we propose EVA (EEG-Vision Alignment), a novel framework that unifies the alignment of multi-scale EEG signals with heterogeneous visual stimuli through contrastive learning with vision-language models. As illustrated in Fig. 1, our framework optimizes EEG feature representations to balance two critical properties: Feature Discriminability and Semantic Fidelity. High discriminability ensures that EEG features from different stimulus categories are well-separated, which is essential for classification and retrieval tasks. High fidelity ensures that EEG features closely match their corresponding visual features, which is crucial for retrieval and reconstruction tasks. This dual optimization is directly inspired by human visual cognition: when viewing stimuli, the brain concurrently performs categorical identification and detailed visual encoding (Clarke & Tyler, 2015). By achieving an optimal balance in the upper-right quadrant of this property space, EVA enables superior performance across diverse neural decoding tasks. Our contributions:

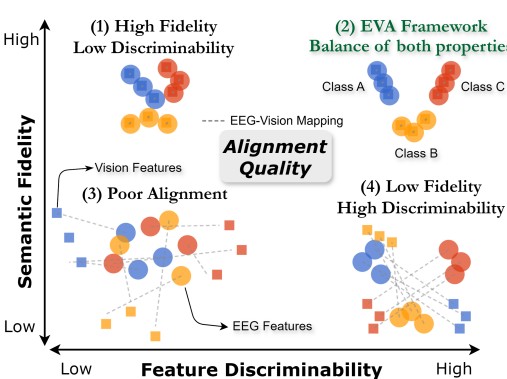

Figure 1: EEG-Vision Alignment Quality Assessment. Circles indicate EEG features, and squares indicate Vision features.

- A Frequency-Aware Dynamic Encoding module that transforms EEG signals into the frequency domain, enabling more compact representation of brain dynamics through adjustable band-pass filtering preserving critical information while controlling compression.
- An Adaptive Channel Clustering module that updates channel groupings through cross-attention mechanisms, featuring real-time adjustment of clustering centers, inter-channel synergy modeling, and end-to-end differentiability.
- The first framework to align multi-scale EEG signals (100ms image presentations, 2s video stimuli, 1s 3D object rotations) with diverse visual modalities, achieving state-of-the-art (SOTA) performance across multiple datasets and tasks.
- Specialized components for alignment, classification, and reconstruction, enabling zero-shot reconstruction of 200 object categories from THINGS-EEG using only aligned EEG features without auxiliary cues, significantly outperforming previous SOTA methods.

## 2 RELATED WORK

**EEG signal encoding models.** EEG encoders are essential for connecting brain signals with vision-language representations. Time-domain approaches like EEG Conformer (Song et al., 2022) com-

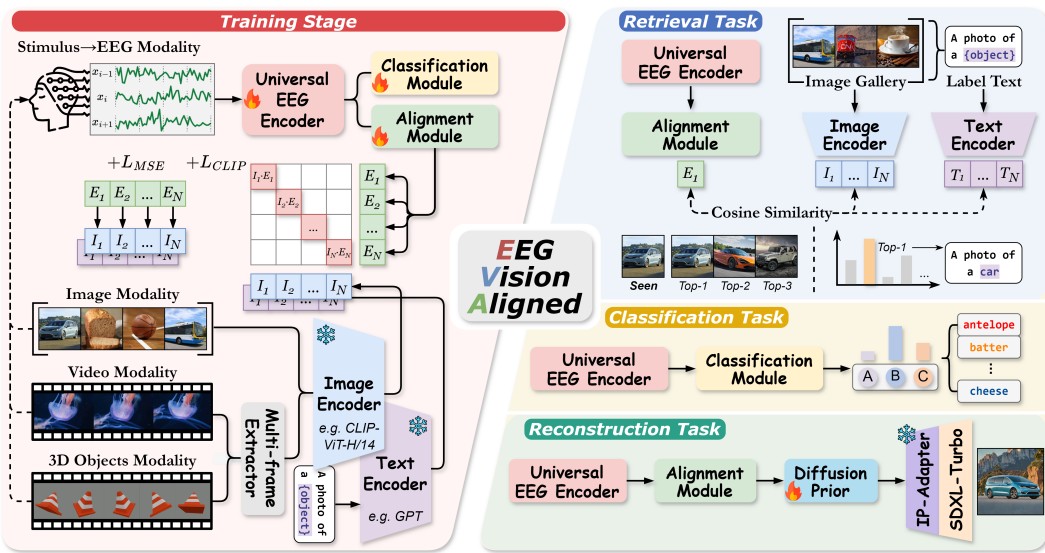

Figure 2: Overview of the EVA architecture and workflow (Training and Inference).

bine CNNs with self-attention (Vaswani et al., 2017) to capture signal patterns, while frequency-domain methods such as MEET (Shi et al., 2023) transform signals into multi-band images preserving spatial electrode relationships. Multi-channel EEG processing has evolved from simple Channel Dependent methods prone to over-smoothing and Channel Independent approaches that ignore inter-channel relationships (Nie et al., 2022), to more sophisticated techniques like DGCNN (Zhang et al., 2020). Recent EEG foundation models including Brant (Zhang et al., 2023a; 2024) leverage masked brain modeling for self-supervised pretraining. However, existing models typically suffer from loss of frequency-specific information, reliance on fixed filter banks, or static channel clustering. Our FADE module addresses these limitations through adaptive frequency-domain processing with adjustable filtering, while our ACC module dynamically optimizes channel groupings via cross-attention and gradient-based learning.

**Brain-vision alignment and reconstruction.** Contrastive learning has transformed multimodal representation learning, with models like CLIP (Radford et al., 2021a) demonstrating powerful zero-shot capabilities by aligning visual and textual representations (Liu et al., 2024a). This approach has advanced brain-to-image reconstruction (Takagi & Nishimoto, 2023), particularly from fMRI data, as seen in MindEye (Scotti et al., 2024; 2023) which maps brain activity to CLIP's image space for high-quality reconstruction. In the EEG domain, NICE (Song et al., 2024) established self-supervised object recognition using contrastive learning, while ATM-S (Li et al., 2024) aligned EEG with CLIP embeddings for image reconstruction. Recent work has expanded beyond static images, with EEG2Video (Liu et al., 2024b) developing a large-scale dataset for video reconstruction from EEG and Neuro-3D (Guo et al., 2024) pioneering 3D visual decoding. Despite this progress, current approaches remain task-specific and struggle to balance discriminative power with semantic fidelity. Our EVA framework addresses these challenges by providing a unified solution for aligning multi-scale EEG signals with diverse visual stimuli while optimizing for both discriminative and semantically faithful representations.

## 3 METHOD

### 3.1 EEG-VISION ALIGNMENT

As illustrated in Fig. 2, EVA establishes a unified framework for aligning multi-scale EEG signals with diverse visual stimuli through contrastive learning. Our framework consists of three main components: (1) a Universal EEG Encoder that transforms raw EEG signals into compact representations, (2) a Classification Module for stimulus category prediction, and (3) an Alignment Module that bridges EEG features with corresponding visual semantics.

During the training stage, EVA processes various visual modalities (images, videos, and 3D objects) and their corresponding EEG recordings. The visual stimuli are encoded through pre-trained vision-language models: CLIP ViT-H/14 (Radford et al., 2021b; Schuhmann et al., 2022; Dosovitskiy et al., 2020) for visual content and GPT (Brown et al., 2020) for textual descriptions, providing high-quality semantic targets. Simultaneously, our Universal EEG Encoder transforms brain signals into a shared representation space optimized for both discriminability and semantic fidelity.

We formulate this dual objective through a joint optimization framework:

$$\mathcal{L} = \alpha\mathcal{L}_{fidelity} + (1 - \alpha)\mathcal{L}_{discrim} + \beta\mathcal{L}_{struct} \tag{1}$$

where $\mathcal{L}_{fidelity}$ encourages EEG features to closely match their corresponding visual features (measured via MSE), $\mathcal{L}_{discrim}$ promotes discriminability between different stimulus categories (implemented through contrastive learning), and $\mathcal{L}_{struct}$ regularizes the latent structure of the EEG representations. The hyperparameters $\alpha$ and $\beta$ control the trade-off between these objectives.

This unified approach enables EVA to generalize across multiple downstream tasks—retrieval, classification, and reconstruction. For retrieval tasks, the Alignment Module computes cosine similarity between EEG features and a gallery of visual or textual embeddings. For classification, the Classification Module directly maps EEG features to stimulus categories. For reconstruction, our framework leverages the aligned EEG features with diffusion priors to generate detailed visual reconstructions of the original stimuli.

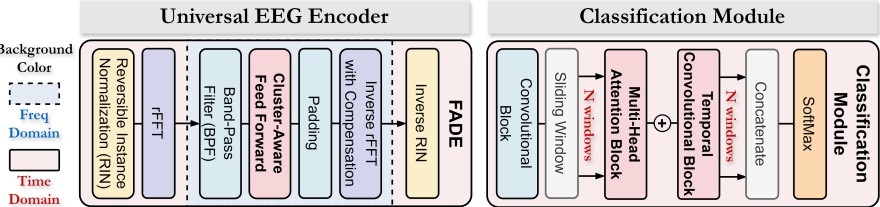

Figure 3: Pipeline of the Universal EEG Encoder and Classification Module.

## 3.2 UNIVERSAL EEG ENCODER

### 3.2.1 FREQUENCY-AWARE DYNAMIC ENCODING (FADE)

The Universal EEG Encoder consists of two key components: Frequency-Aware Dynamic Encoding and Adaptive Channel Clustering. EEG signals elicited during rapid visual stimulation (on the millisecond scale) exhibit complex temporal dynamics that are challenging to model directly in the time domain. Traditional approaches using recurrent or convolutional architectures often struggle to efficiently capture the relevant frequency patterns. FADE addresses this challenge by leveraging frequency domain transformations to extract compact and informative representations. The module operates through the following process, as shown in Fig. 3:

**Frequency transformation**: We convert time-domain EEG signals to the frequency domain using the real Fast Fourier Transform (rFFT): $\hat{X} = \text{rFFT}(X) \in \mathbb{C}^{C \times F}$, where $C$ is the number of EEG channels and $F$ represents the number of frequency components.

**Spectral processing**: FADE incorporates an adjustable bandpass filtering mechanism to focus on relevant frequency bands while eliminating extraneous components. This step not only reduces noise but also compresses the representation while preserving essential EEG characteristics. The filtered frequency representation is then processed through a channel-wise spectral encoder that captures frequency-specific patterns.

**Inverse mapping**: The processed frequency representation is mapped back to the time domain using the inverse real Fast Fourier Transform (irFFT): $Z = \text{irFFT}(\hat{Z}) \in \mathbb{R}^{C \times T'}$, where $T'$ may differ from the original signal length, requiring zero-padding prior to the inverse transformation.

This approach is effective for rapid visual stimuli where transient neural responses may be obscured by noise in the time domain but can be effectively isolated in the frequency domain. The FADE module enables our model to maintain semantic coherence across different temporal scales, from brief image presentations (100ms) to extended video sequences (2s) and 3D object rotations (1s).

### 3.2.2 ADAPTIVE CHANNEL CLUSTERING (ACC)

ACC dynamically groups EEG channels based on their functional relationships rather than fixed anatomical positions. It features three key innovations:

**Dynamic cluster centers and soft cluster assignment**: The module initializes $K$ learnable cluster embeddings $c_1, ..., c_K$, where each $c_k \in \mathbb{R}^d$ (with $d$ representing the hidden dimension), that adapt to the evolving distribution of channel features during training. Given an EEG input $X$, each channel is transformed into a $d$-dimensional embedding $h_i$ using a linear projection. The association between channel $i$ and cluster $k$ is determined by computing a probability:

$$p_{i,k} = \text{Normalize}(\frac{c_k^\top h_i}{\|c_k\|\|h_i\|}) \in [0, 1] \tag{2}$$

**Cluster updating via cross-attention**: We employ a mask-based cross-attention mechanism to update the cluster embeddings based on channel features:

$$\hat{\mathbf{C}} = \text{Normalize}\left(\exp(\frac{(W_Q\mathbf{C})(W_K\mathbf{H})^\top}{\sqrt{d}}) \odot \mathbf{M}^\top\right) W_V\mathbf{H} \tag{3}$$

where $\mathbf{C} = [c_1, ..., c_K] \in \mathbb{R}^{K \times d}$ is the cluster embedding matrix, $\mathbf{H} = [h_1, ..., h_C] \in \mathbb{R}^{C \times d}$ is the channel embedding matrix, and $W_Q$, $W_K$, and $W_V$ are learnable weight matrices. The mask matrix $\mathbf{M}$ is derived using a reparameterization technique to approximate a Bernoulli distribution.

**Differentiable optimization**: To enable end-to-end training, we introduce a spectral clustering-inspired regularization term:

$$\mathcal{L}_{struct} = -\text{Tr}(\tilde{\mathbf{P}}^\top\mathbf{S}\tilde{\mathbf{P}}) + \text{Tr}\left((\mathbf{I} - \tilde{\mathbf{P}}\tilde{\mathbf{P}}^\top)\mathbf{S}\right) + \lambda \sum_{c,k} -\mathbf{P}_{ck}\log\mathbf{P}_{ck} \tag{4}$$

where $\tilde{\mathbf{P}}$ is a softened assignment matrix derived using Gumbel-Softmax relaxation, and $\mathbf{S}$ denotes the channel similarity matrix. The first term maximizes similarities within clusters, the second penalizes similarities between different clusters, and the entropy term prevents trivial solutions where all channels collapse into a single cluster. The implementation of ACC is relatively complex, further details are provided in Appendix A.1.

### 3.3 TASK-SPECIFIC MODULE

To adapt our Universal EEG Encoder for various downstream applications, we develop specialized modules for classification, alignment, and reconstruction tasks.

**Classification module.** To enhance the classification performance by integrating both frequency and time domain information, we have fine-tuned the structure of Universal EEG Encoder and simplified the computation process as follows. The rationale for this design is provided in the Appendix A.8.

$$\text{FreqEnhanced}(X) = X \odot W_f + \alpha \cdot \mathcal{F}^{-1}(|F| \odot M \cdot e^\phi) \tag{5}$$

where $W_f$ represents learnable channel weights, $F = \mathcal{F}(X)$ is the rFFT of input signal $X$, $|F|$ and $\phi$ are the magnitude and phase components, $\mathcal{F}^{-1}$ denotes the irFFT, $M$ is a frequency mask emphasizing bands below a dominance threshold, and $\alpha$ controls fusion intensity.

As shown in Fig. 3, the enhanced signals flow through a multi-stage pipeline: (1) A convolutional block with temporal and spatial filters extracts local patterns; (2) A sliding window approach divides features into overlapping segments, with each processed by self-attention and temporal convolutional networks; (3) Finally, features from all windows are concatenated and mapped to classification logits through max-norm constrained linear layers. This multi-faceted architecture effectively captures frequency characteristics, spatial relationships between channels, and temporal dependencies at multiple scales, yielding discriminative features for accurate stimulus classification.

**Alignment module and reconstruction pipeline.** The alignment module is designed to map EEG features to the same semantic space as visual features extracted from vision-language models. Inspired by ShallowNet (Bai et al., 2018), we implement a Spatial-Temporal ConvNet that effectively captures both spatial and temporal patterns in EEG features (He et al., 2016). This module employs a

contrastive learning approach to optimize both discriminability and semantic fidelity of the extracted features.

For image reconstruction, our framework offers a streamlined and efficient pipeline. Using only the EEG feature $z_e$ encoded by the Universal EEG Encoder and alignment module, we aim to obtain image features $z_i$ through a prior diffusion model. Assuming the feature output from the prior is $z_i'$, our training objective is to minimize the distance between $z_i'$ and $z_i$. The fully trained prior is then integrated with IP-Adapter (Ye et al., 2023) and Stable Diffusion (SDXL-Turbo) (Podell et al., 2023) to achieve high-quality reconstruction of stimulus images.

# 4 EXPERIMENTS AND RESULTS

## 4.1 DATASETS AND EXPERIMENTAL DESIGN

**Datasets.** We evaluated our framework across three distinct visual modalities using complementary datasets. For EEG-Image alignment, the THINGS-EEG dataset (Gifford et al., 2022) provided high temporal resolution EEG responses from 10 participants viewing 16,740 unique image conditions, totaling 82,160 trials per participant presented via rapid serial visual presentation (RSVP) (Potter, 2018). For EEG-Video alignment, the SEED-DV dataset (Liu et al., 2024b) comprised EEG signals from 20 subjects watching 1,400 dynamic video clips (2 seconds each) spanning 40 conceptual categories. For EEG-3D object alignment, the EEG-3D dataset (Guo et al., 2024) contained EEG recordings from 12 subjects viewing 72 categories of 3D objects,

**Experimental design.** We benchmarked against multiple SOTA methods across various domains: (1) EEG-specific encoders: TSConv (Song et al., 2024), EEG Conformer (Song et al., 2022), ShallowFBCSPNet (Schirrmeister et al., 2017), EEGNet (Lawhern et al., 2018), EEGNetV4 (Lawhern et al., 2018), DeepNet (Schirrmeister et al., 2017); (2) EEG foundation models: BrainBERT (Wang et al., 2023), Neuro-GPT (Cui et al., 2024), FoME (Shi et al., 2024), LaBraM (Jiang et al., 2024), CBraMod (Wang et al., 2024); (3) temporal models: PatchTST (Nie et al., 2022), DLinear (Zeng et al., 2023); and (4) EEG-Vision alignment models: NICE (Song et al., 2024), ATM-S (Li et al., 2024), EEG2Video (Liu et al., 2024b), Neuro-3D (Guo et al., 2024). All experiments were conducted using PyTorch 2.1.2 with NVIDIA RTX 4090 GPUs and CUDA 12.4.

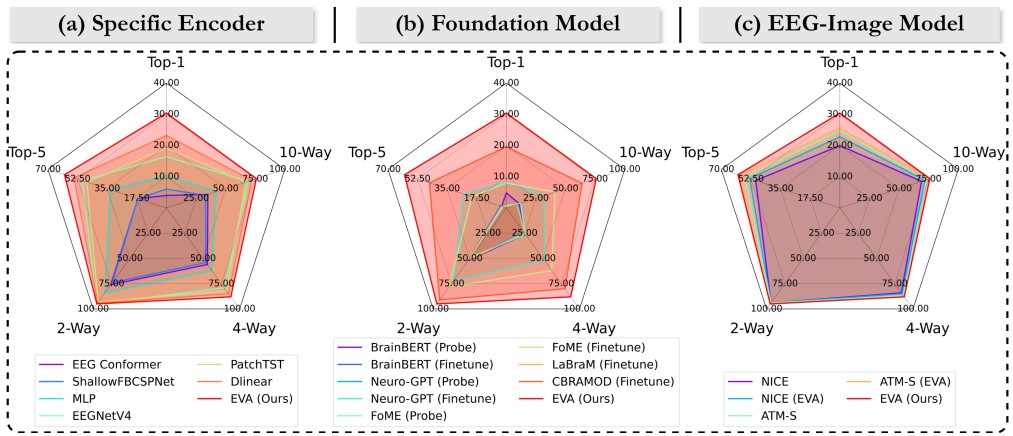

Figure 4: Zero-shot image retrieval performance (accuracy %) on the THINGS-EEG dataset. See Appendix A.4 for additional results.

## 4.2 EVALUATING FEATURE DISCRIMINABILITY OF EVA

We assessed the discriminative capacity of EVA-generated EEG features through multiple cross-modal retrieval and classification tasks.

**Image retrieval.** Using the THINGS-EEG dataset, we evaluated the framework's ability to retrieve the correct visual stimulus from a pool of 200 candidates based on EEG representations. As pre-

sented in Fig. 4, EVA achieved 30.55% Top-1 accuracy and 59.90% Top-5 accuracy, substantially outperforming leading alternatives including ATM-S (24.70% Top-1, 55.90% Top-5) and DLinear (23.25% Top-1, 54.70% Top-5). Notably, EVA's Top-1 accuracy exceeded ATM-S by 5.85 percentage points and DLinear by 7.3 percentage points, establishing a new benchmark for this task. These results indicate that EEG features encoded by EVA are sufficiently distinct and discriminative to accurately identify their corresponding visual counterparts from a large set of candidates.

Table 1: Video classification performance (accuracy %) evaluated across all subjects. See Appendix A.5 for additional results.

| Year | Method | Top-1 | Top-5 | Color | Face | Human | N. Obj | F / S |
|---|---|---|---|---|---|---|---|---|
| | Chance level | 2.50 | 12.50 | 20.57 | 62.25 | 71.43 | 65.64 | 50.00 |
| 1986 | MLP | 5.48 | 18.28 | 21.32 | 69.24 | 69.49 | 62.61 | 52.68 |
| 2017 | ShallowFBCSPNet | 6.01 | 19.82 | 23.75 | 72.54 | 71.01 | 60.47 | 53.71 |
| 2017 | DeepNet | 4.56 | 14.30 | 26.37 | 61.58 | 72.86 | 65.71 | 55.42 |
| 2018 | EEGNet | 4.64 | 14.25 | 25.46 | 61.37 | 72.38 | 64.67 | 51.99 |
| 2018 | EEGNetv4 | 6.48 | 20.73 | 24.72 | 74.91 | 70.38 | 63.46 | 51.17 |
| 2022 | EEG Conformer | 4.93 | 15.36 | 27.53 | 64.96 | 73.00 | 65.73 | 55.02 |
| 2023 | DLinear | 5.56 | 18.20 | 21.33 | 68.44 | 70.09 | 61.80 | 53.13 |
| 2024 | TSConv | 4.92 | 15.05 | 26.89 | 64.39 | 72.68 | 65.39 | 55.32 |
| 2024 | GLMNet (EEG2Video) | 6.20 | 17.75 | 27.33 | 65.10 | 73.34 | 66.21 | 57.35 |
| Ours | EVA (Alignment Module) | 6.53 | 19.50 | 22.01 | 70.52 | 71.05 | 59.45 | 53.64 |
| | EVA (Ours) | **7.88** | **24.01** | **31.50** | **78.22** | **73.66** | **68.53** | **57.65** |

**Video classification.** On the SEED-DV dataset, we assessed the classification of 40 semantic categories from EEG signals. As shown in Table 1, EVA attained 7.88% Top-1 and 24.01% Top-5 accuracy, surpassing EEGNetv4 (6.48% Top-1) and EEG2Video (6.20% Top-1). The framework also demonstrated robust performance on meta-information classification tasks: color (31.50%), face detection (78.22%), and human presence (73.66%). These results indicate EVA's efficacy in capturing discriminative temporal patterns from EEG signals related to complex video stimuli.

Table 2: Performance on 3D visual classification and retrieval tasks using the EEG-3D dataset.

| Task | Year | Method | Object Type | | Color Type | |
|---|---|---|---|---|---|---|
| | | | Top-1 | Top-5 | Top-1 | Top-2 |
| | | Chance level | 1.39 | 6.94 | 16.67 | 33.33 |
| Classification | 2017 | DeepNet | 3.70 | 9.90 | 20.95 | 49.71 |
| | 2018 | EEGNet | 3.82 | 9.72 | 18.35 | 46.47 |
| | 2022 | EEG Conformer | 4.05 | 10.30 | 18.27 | 35.81 |
| | 2024 | TSConv | 4.05 | 10.13 | 31.13 | 59.49 |
| | 2024 | Neuro-3D | 5.91 | **16.30** | 39.93 | 61.40 |
| | Ours | EVA | **6.11** | 16.25 | **40.70** | **63.61** |
| Retrieval | 2024 | Neuro-3D | 5.42 | 16.25 | –.– | –.– |
| | Ours | EVA | **5.70** | **16.39** | –.– | –.– |

**3D object classification and retrieval.** On the EEG-3D dataset, as detailed in Table 2, EVA achieved 6.11% Top-1 accuracy for object type classification across 72 categories, marginally exceeding Neuro-3D's performance (5.91% Top-1). For color classification across 6 types, EVA demonstrated 40.70% Top-1 and 63.61% Top-2 accuracy, compared to Neuro-3D's 39.93% and 61.40%, respectively. In retrieval tasks, EVA attained 5.70% Top-1 and 16.39% Top-5 accuracy, again showing incremental improvements over Neuro-3D (5.42% Top-1, 16.25% Top-5). These consistent improvements over a specialized 3D decoding model underscore EVA's versatility and its ability to generate discriminative features for complex 3D visual stimuli.

Collectively, these findings across varied visual modalities and tasks provide compelling evidence for the discriminability of EVA-generated EEG features, a critical prerequisite for neural decoding.

### 4.3 EVALUATING SEMANTIC FIDELITY OF EVA

Beyond discriminability, we evaluated the semantic fidelity of our framework—the degree to which encoded EEG features preserve the semantic essence of corresponding visual stimuli.

**Multi-frame alignment.** We quantified alignment fidelity on the SEED-DV dataset by measuring Mean Squared Error (MSE) between EVA-encoded EEG features and corresponding image features extracted from video frames using three pre-trained visual encoders: OpenCLIP-ViT-H/14 (Radford et al., 2021b; Schuhmann et al., 2022), SigLIP-Large-patch16-256 (Zhai et al., 2023), and DINOv2-Large (Oquab et al., 2023). As illustrated in Fig. 5, EVA consistently achieved the lowest MSE across all visual encoders and sequence lengths compared to alternative approaches (EEGNetv4, MLP, NICE-Encoder). This advantage became more pronounced with increasing sequence length, demonstrating EVA's capacity to map continuous EEG signals to evolving video content with superior fidelity.

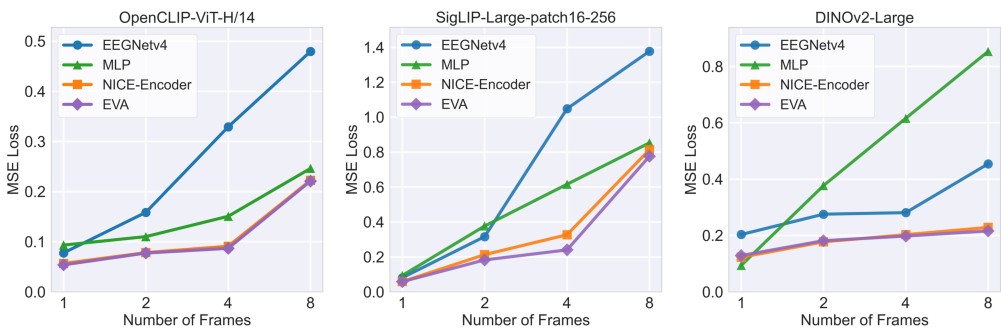

Figure 5: Multi-frame alignment performance (MSE loss) on the SEED-DV dataset.

**Zero-shot image reconstruction.** Further, to demonstrate the practical implications of high semantic fidelity, we evaluated EVA on zero-shot image reconstruction using the THINGS-EEG dataset. This challenging task required generating recognizable images from EEG signals without specific image-EEG pair training. Crucially, our reconstruction pipeline utilized only EVA-aligned EEG features, without auxiliary information such as text prompts or low-level visual features often employed in other methods. As shown in Fig. 6, EVA-derived features produced reconstructions that more accurately captured key semantic elements, shapes, colors, and textures of the original stimuli compared to ATM-S (Li et al., 2024). EVA reconstructions demonstrated clearer object forms, more appropriate color palettes, and better overall resemblance to ground truth images across both simple and complex visual scenes.

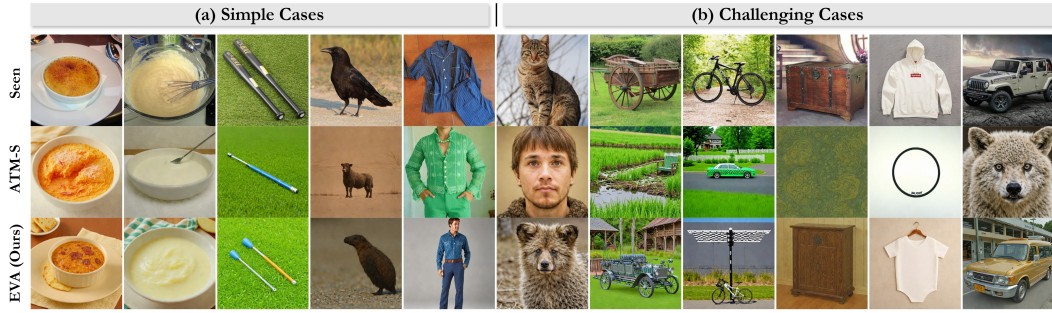

Figure 6: Qualitative results of zero-shot image reconstruction from the THINGS-EEG dataset. See Appendix A.6 for more cases.

These results substantiate EVA's capacity to encode EEG features with high semantic fidelity, which, combined with the discriminability demonstrated earlier, underpins the framework's robust performance across diverse neural decoding applications.

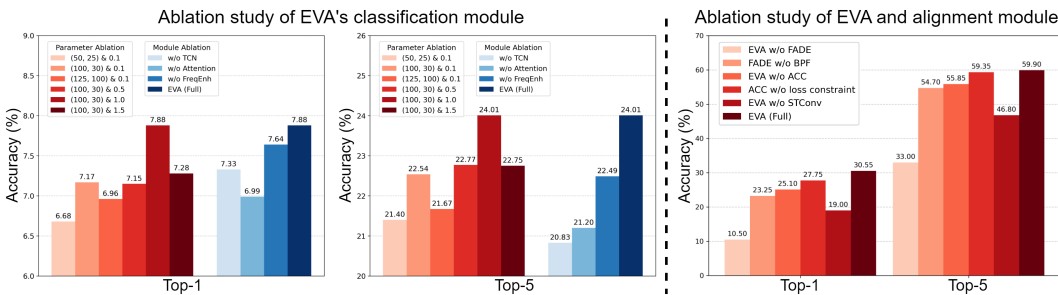

Figure 7: Ablation study results on the SEED-DV (left) and THINGS-EEG (right) datasets. See Appendix A.5 for additional results.

## 4.4 ABLATION STUDIES AND INTERPRETABILITY ANALYSIS

To validate the contributions of our architectural choices, we conducted ablation studies on the THINGS-EEG and SEED-DV datasets, with results shown in Fig. 7. These studies collectively affirm that the proposed FADE and ACC modules, along with the carefully designed classification head, are integral to EVA's state-of-the-art performance.

Visualization of ACC's cluster-wise linear layer weights (Fig. 8) revealed distinct processing strategies across clusters. For instance, while Cluster 2 exhibited diffuse weight patterns suggesting global feature processing, Cluster 4 displayed highly localized and pronounced positive/negative weights, indicating specialized selective emphasis of specific input features.

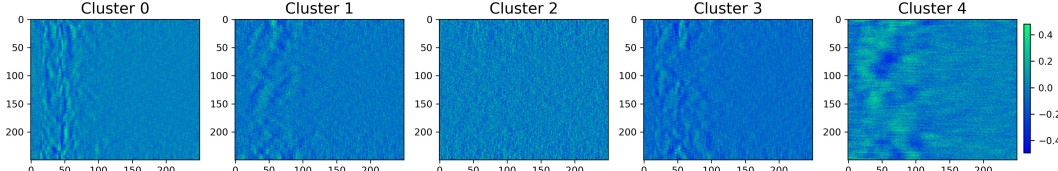

Figure 8: Visualization of learned weights for cluster-wise linear layers within the ACC module.

## 5 CONCLUSION AND DISCUSSION

**Conclusion.** In this work, we presented EVA, a novel framework for aligning multi-scale EEG signals with diverse visual stimuli through contrastive learning. By introducing the FADE module for domain transformation and the ACC module for dynamic channel grouping, our approach effectively balances feature discriminability and semantic fidelity. The theoretical foundation for this dual optimization is detailed in Appendix A.2. Extensive experiments across multiple datasets demonstrated EVA's superior performance in various neural decoding tasks, including image retrieval, video classification, and 3D object recognition. Most notably, our framework enabled zero-shot reconstruction from the THINGS-EEG dataset using only aligned EEG features, substantially outperforming previous SOTA methods. These results highlight EVA's ability to extract robust, generalizable representations from complex EEG signals, advancing the field of cross-modal neural decoding.

**Limitations and future works.** Despite EVA's promising results, several limitations remain. First, the framework's performance may vary across individuals due to neurophysiological differences, suggesting the need for personalization strategies. Second, while our approach handles diverse visual stimuli, extending it to other sensory modalities (e.g., auditory, tactile) would provide a more comprehensive neural decoding solution. Future work should explore online adaptation techniques to accommodate neural plasticity and investigate transfer learning capabilities across datasets and tasks. Building upon the solid reconstruction pipeline established by EVA, incorporating textual semantic information, low-level features, and deep representations to control structural elements and refine details could substantially enhance reconstruction quality.

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

# A  APPENDIX

## A.1  THE DETAILED IMPLEMENTATION OF ADAPTIVE CHANNEL CLUSTERING

EEG signals are captured across multiple electrode channels distributed over the scalp, each recording activity from different brain regions. Existing approaches for multi-channel processing typically adopt one of three strategies: (1) channel-dependent approaches that mix all channels indiscriminately, risking over-smoothing; (2) channel-independent approaches that process each channel separately, neglecting inter-channel relationships; or (3) prior knowledge-based approaches that group channels according to fixed anatomical regions. Some recent methods employ hard clustering techniques to group channels, but these assignments remain static throughout training, limiting adaptability.

To overcome these limitations, we propose the Adaptive Channel Clustering (ACC) module, which dynamically groups EEG channels based on their functional relationships rather than fixed anatomical positions.

### A.1.1  CHANNEL CLUSTERING WITH LEARNABLE QUERIES

ACC adaptively learns channel groupings through a fully differentiable clustering mechanism, enabling end-to-end optimization within our framework. The algorithm proceeds as follows:

**Initialization**: We initialize $K$ learnable cluster embeddings $c_1, ..., c_K$, where each $c_k \in \mathbb{R}^d$ (with $d$ representing the hidden dimension). These cluster centers serve as prototype representations for different functional channel groups.

**Channel embedding**: Given an EEG input $X \in \mathbb{R}^{C \times T}$ with $C$ channels and $T$ time points, we transform each channel into a $d$-dimensional embedding using a linear projection: $h_i = W_h \cdot X_i + b_h$ where $X_i \in \mathbb{R}^T$ represents the time series of the $i$-th channel, and $W_h \in \mathbb{R}^{d \times T}$ and $b_h \in \mathbb{R}^d$ are learnable parameters.

**Soft cluster assignment**: The association between channel $i$ and cluster $k$ is determined by computing a normalized similarity score:

$$p_{i,k} = \frac{\exp\left(\frac{c_k^\top h_i}{|c_k||h_i| \cdot \tau}\right)}{\sum_{j=1}^{K} \exp\left(\frac{c_j^\top h_i}{|c_j||h_i| \cdot \tau}\right)} \in [0, 1] \tag{6}$$

where $\tau$ is a temperature parameter controlling the softness of the assignment. This creates a probability distribution over the $K$ clusters for each channel.

**Mask generation**: To enable gradient-based optimization while approximating discrete assignments, we apply a Gumbel-Softmax reparameterization technique:

$$\tilde{p}_{i,k} = \frac{\exp\left((\log p_{i,k} + g_{i,k})/\gamma\right)}{\sum_{j=1}^{K} \exp\left((\log p_{i,j} + g_{i,j})/\gamma\right)} \tag{7}$$

where $g_{i,k} \sim \text{Gumbel}(0, 1)$ are random samples from the Gumbel distribution, and $\gamma$ is an annealing temperature parameter. From these probabilities, we derive the clustering mask matrix $\mathbf{M} \in \mathbb{R}^{C \times K}$, where each element $M_{i,k}$ approximates a Bernoulli distribution. Higher probability values $p_{i,k}$ translate to $M_{i,k}$ values closer to 1, indicating strong association between channel $i$ and cluster $k$.

### A.1.2  CLUSTER UPDATING VIA CROSS-ATTENTION

A key innovation in ACC is the dynamic updating of cluster prototypes based on the current channel features and their cluster assignments. We implement this through a mask-based cross-attention mechanism:

**Query, Key, Value projection**: We define the cluster embedding matrix $\mathbf{C} = [c_1, ..., c_K] \in \mathbb{R}^{K \times d}$ and the channel embedding matrix $\mathbf{H} = [h_1, ..., h_C] \in \mathbb{R}^{C \times d}$. We project these matrices to obtain query, key, and value representations:

$$\mathbf{Q} = W_Q \mathbf{C}, \quad \mathbf{K} = W_K \mathbf{H}, \quad \mathbf{V} = W_V \mathbf{H} \tag{8}$$

where $W_Q, W_K, W_V \in \mathbb{R}^{d \times d}$ are learnable parameter matrices.

**Attention with mask**: We compute the attention scores between clusters and channels, and use the transpose of the mask matrix $\mathbf{M}^\mathsf{T}$ to focus attention on relevant channel-cluster pairs:

$$\mathbf{A} = \text{Softmax}\left(\frac{\mathbf{Q}\mathbf{K}^\mathsf{T}}{\sqrt{d}} \odot \mathbf{M}^\mathsf{T}\right) \tag{9}$$

where $\odot$ represents element-wise multiplication. This ensures that each cluster primarily attends to channels that have been assigned to it.

**Cluster update**: The refined cluster embeddings are computed as:

$$\hat{\mathbf{C}} = \mathbf{A}\mathbf{V} \tag{10}$$

These updated prototypes capture the shared patterns of channels within each cluster, adapting to the evolving features during training.

### A.1.3 Cluster-aware Feed-Forward Networks

Rather than processing all channels with the same weights or using separate weights for each channel, we utilize the soft cluster assignments to share parameters efficiently:

**Cluster-specific processing**: For each cluster $k$, we define a separate feed-forward network $f_{\theta_k}(\cdot)$ with parameters $\theta_k$.

**Weighted processing**: Given a channel embedding $z_i$, its processed representation is computed as the weighted combination of outputs from all cluster networks:

$$\hat{z}_i = \sum_{k=1}^{K} p_{i,k} \cdot f_{\theta_k}(z_i) \tag{11}$$

where $p_{i,k}$ is the assignment probability of channel $i$ to cluster $k$.

This approach allows channels with similar functional properties to share parameters, while still accounting for their unique characteristics through the soft assignment weights.

### A.1.4 Differentiable Cluster Optimization

To guide the learning of meaningful channel groupings, we introduce a spectral clustering-inspired regularization term:

$$\mathcal{L}_{\text{struct}} = -\text{Tr}(\tilde{\mathbf{P}}^\mathsf{T}\mathbf{S}\tilde{\mathbf{P}}) + \text{Tr}\left((\mathbf{I} - \tilde{\mathbf{P}}\tilde{\mathbf{P}}^\mathsf{T})\mathbf{S}\right) + \lambda \sum_{c=1}^{C}\sum_{k=1}^{K} -\mathbf{P}_{c,k}\log\mathbf{P}_{c,k} \tag{12}$$

This loss function consists of three components:

1. Intra-cluster Similarity: The term $-\text{Tr}(\tilde{\mathbf{P}}^\mathsf{T}\mathbf{S}\tilde{\mathbf{P}})$ encourages channels within the same cluster to have high similarity, where $\mathbf{S} \in \mathbb{R}^{C \times C}$ is the channel similarity matrix computed based on feature correlations.

2. Inter-cluster Dissimilarity: The term $\text{Tr}\left((\mathbf{I} - \tilde{\mathbf{P}}\tilde{\mathbf{P}}^\mathsf{T})\mathbf{S}\right)$ penalizes high similarity between channels assigned to different clusters.

3. Entropy Regularization: The term $\lambda \sum_{c=1}^{C}\sum_{k=1}^{K} -\mathbf{P}_{c,k}\log\mathbf{P}_{c,k}$ with hyperparameter $\lambda$ prevents trivial solutions where all channels collapse into a single cluster or each channel forms its own cluster.

### A.1.5 Algorithm Summary

The complete ACC algorithm can be summarized as Algorithm 1:

---

**Algorithm 1:** Adaptive Channel Clustering (ACC)

---

**Input:** EEG input $X \in \mathbb{R}^{C \times T}$, number of clusters $K$, temperature $\tau$, annealing parameter $\gamma$

**Output:** Updated cluster embeddings $\hat{\mathbf{C}}$, processed channel features $\{\hat{z}_i\}_{i=1}^{C}$, structural loss $\mathcal{L}_{\text{struct}}$

1 Initialize learnable cluster centers $\{c_1, ..., c_K\}$ where $c_k \in \mathbb{R}^d$

2 Project each channel: $h_i = W_h \cdot X_i + b_h$ for $i = 1, ..., C$

3 **for** *each channel $i$ and cluster $k$* **do**

4      Compute soft assignment: $p_{i,k} = \dfrac{\exp\left(\frac{c_k^\top h_i}{|c_k||h_i| \cdot \tau}\right)}{\sum_{j=1}^{K} \exp\left(\frac{c_j^\top h_i}{|c_j||h_i| \cdot \tau}\right)}$

5 **end**

6 Sample Gumbel noise: $g_{i,k} \sim \text{Gumbel}(0,1)$ for all $i, k$

7 Generate mask matrix: $\mathbf{M}_{i,k} = \dfrac{\exp((\log p_{i,k} + g_{i,k})/\gamma)}{\sum_{j=1}^{K} \exp((\log p_{i,j} + g_{i,j})/\gamma)}$

8 Compute attention with mask: $\mathbf{A} = \text{Softmax}\left(\frac{(W_Q\mathbf{C})(W_K\mathbf{H})^\top}{\sqrt{d}} \odot \mathbf{M}^\top\right)$

9 Update cluster prototypes: $\hat{\mathbf{C}} = \mathbf{A}(W_V\mathbf{H})$

10 **for** *each channel $i$* **do**

11      Process with cluster-aware FFN: $\hat{z}_i = \sum_{k=1}^{K} p_{i,k} \cdot f_{\theta_k}(z_i)$

12 **end**

13 Compute structural loss: $\mathcal{L}_{\text{struct}} = -\text{Tr}(\tilde{\mathbf{P}}^\top \mathbf{S}\tilde{\mathbf{P}}) + \text{Tr}((\mathbf{I} - \tilde{\mathbf{P}}\tilde{\mathbf{P}}^\top)\mathbf{S}) + \lambda \sum_{c,k} -\mathbf{P}_{c,k} \log \mathbf{P}_{c,k}$

14 **return** $\hat{\mathbf{C}}, \{\hat{z}_i\}_{i=1}^{C}, \mathcal{L}_{\text{struct}}$

---

By dynamically adapting to the functional properties of EEG channels rather than relying on fixed anatomical groupings, ACC enables more effective feature extraction than traditional approaches. This is particularly important for EEG-based visual semantic decoding, where relevant neural patterns may span multiple brain regions and evolve differently across subjects, tasks, and stimulus types.

A.2  The Connection Between Brain-Inspired Motivation and Methodology

Our framework's core principle of optimizing for both discriminability and semantic fidelity is not merely a technical choice but a fundamental design philosophy inspired by human visual cognition. This section provides detailed justification for this approach and demonstrates its empirical validity.

A.2.1  Motivation Drives Methodology

The principle of balancing discriminability and fidelity directly informed our choice of a joint-optimization framework with distinct loss terms. A project focused solely on classification would likely exclude generative losses (MSE to visual priors), while a reconstruction-only approach might neglect contrastive terms essential for class separation. Inspired by the brain's dual capabilities (Clarke & Tyler, 2015; Chen et al., 2017), our goal was to create a single versatile encoder capable of supporting both outcomes simultaneously.

We acknowledge that optimally balancing these objectives represents a complex challenge, and while our multi-loss approach constitutes a significant advance, it may not represent the final solution. To demonstrate that these properties are deeply intertwined and that our motivation directly connects to our experimental findings, we present two concrete examples from our results:

**Case 1: NICE vs. EVA Performance Analysis.** During training, the NICE encoder achieves an MSE loss of 0.07841 when aligning EEG embeddings to CLIP embeddings, remarkably close to EVA's 0.07748. This similarity suggests comparable initial alignment fidelity between feature pairs. However, downstream performance reveals stark differences: NICE's reconstruction quality is substantially lower (Table 9), and its retrieval accuracy reaches only 20.08% compared to EVA's 30.55%. This demonstrates that similar initial alignment fidelity becomes insufficient when discriminability is poor, severely compromising final reconstruction quality and proving the interdependence of these properties.

**Case 2: ATM vs. EVA Reconstruction Analysis.** In Figure 6, ATM baseline produces reconstructions that match the style and color palette of original stimuli (columns 6 'cat', 7 'cart', 8 'bike', 11 'jeep'), suggesting high global feature fidelity. However, the core objects in these reconstructions are incorrect. EVA correctly reconstructs the primary objects in these cases. This illustrates the opposite effect: high stylistic fidelity proves insufficient when poor discriminability prevents correct object identification, resulting in semantically flawed outputs.

Both examples demonstrate that discriminability and fidelity function as mutually supportive properties rather than independent objectives. Our core motivation of explicitly balancing these properties therefore represents a necessary principle requiring both effective loss structure and, fundamentally, a powerful encoder like EVA to provide high-quality features enabling such balance.

### A.2.2 MOTIVATION DRIVES ENCODER DESIGN

This principle directly informed our encoder architecture. To create embeddings sufficiently rich to support both discriminative and reconstructive tasks, representations must be both noise-free and compact. This requirement for high-quality feature extraction motivated the development of FADE (spectral noise reduction and compact frequency representations) and ACC (efficient channel-wise feature compression). Our ablation studies confirm that removing these components degrades performance on both task types, demonstrating the encoder's intrinsic connection to our central goal of creating balanced, versatile representations.

### A.2.3 CONCEPTUAL CONTRIBUTION

This framing represents a conceptual contribution to the field. While prior work employed similar losses as technical tools, we explicitly posit that balancing discriminability and fidelity constitutes a fundamental objective for future general-purpose brain-computer interfaces. This framework guides research toward creating more holistic and capable neurotechnologies that mirror the brain's own dual processing capabilities.

### A.3 DATASET DESCRIPTIONS

**THINGS-EEG dataset.** The THINGS-EEG dataset, utilized for EEG-Image alignment tasks, was developed to model the dynamics of human visual object recognition using high-resolution EEG. Data were collected from 10 healthy adults who viewed images from the THINGS database (Hebart et al., 2023), depicting objects on natural backgrounds. The study employed a rapid serial visual presentation (RSVP) paradigm where each image was shown for 100 ms with a 200 ms stimulus onset asynchrony, while participants performed an orthogonal target detection task. The dataset is extensive, containing 1,854 object concepts split into 1,654 training concepts (10 images per concept, 16,540 unique training images, each repeated 4 times) and 200 test concepts (1 image per concept, 200 unique test images, each repeated 80 times), totaling 82,160 trials per participant. EEG data were recorded from a 64-channel EASYCAP system at a 1000 Hz sampling rate, online filtered (0.1-100 Hz), and later epoched from -200 ms to 800 ms relative to stimulus onset.

**SEED-DV dataset.** For EEG-Video alignment, the SEED-DV dataset was employed. This dataset was created to facilitate research into decoding dynamic visual perception by providing EEG-video paired data. It includes EEG signals from 20 healthy student participants (10 females, 10 males, mean age: 21.75) while they watched 1,400 two-second dynamic video clips. These clips represented 40 distinct concepts, which were also grouped into 9 coarser classes, with 35 unique video clips available for each fine-grained concept. The experimental paradigm involved presenting videos in 7 blocks, each block comprising 200 clips (5 clips for each of the 40 concepts presented in a randomized order per block). A 3-second hint preceded each group of 5 same-class videos, and each block lasted approximately 8 minutes and 40 seconds, with at least a 30-second rest between blocks. EEG data were acquired using a 62-channel AgCl electrode cap (10-10 system) with an ESI NeuroScan System at a 1000 Hz sampling rate. Preprocessing involved a 0.1-100 Hz band-pass filter and down-sampling to 200 Hz. For EEG segmentation, the Multi-frame Extractor uses four non-overlapping sliding windows of different sizes (2s, 1s, 500ms, and 250ms) to obtain signals at four different scales, corresponding to the extraction of 1, 2, 4, and 8 stimulus frames, respectively. Data splitting for classification involved 7-fold cross-validation (One block for testing, the previous one for validation, and the rest for training).

**EEG-3D dataset.** The EEG-3D dataset, used for EEG-3D object alignment, offers paired EEG signals with 3D object stimuli to investigate the neural basis of 3D visual perception. It contains extensive EEG recordings from 12 healthy adult participants (5 males, 7 females, mean age: 21.08) who viewed 72 categories of 3D objects sourced from the Objaverse dataset (10 objects per category). The visual stimuli comprised both 6-second rotating videos (30 Hz) of the 3D objects and 0.5-second static images (the initial and final frames of these videos). Each stimulus block presented a static image, then the rotating video, followed by a static image, with blank screens and a 1-second fixation cross between object blocks. Objects designated for training received 2 measurement repetitions, whereas test set objects received 4 repetitions, conducted over 24 sessions for each participant, totaling approximately 5.5 hours of experiment time per participant, including 5-minute resting-state EEG recordings at the beginning and end of all sessions. EEG data were recorded from a 64-channel EASYCAP system (10-10 system) at 1000 Hz. Preprocessing included segmenting the continuous EEG into 1s epochs for static stimuli and 6s epochs for dynamic stimuli, down-sampling to 250 Hz, applying a 0.1-100 Hz bandpass filter and a 50 Hz notch filter, and performing multivariate noise normalization.

Table 3: Evaluation results (accuracy %) of zero-shot retrieval task based on THINGS-EEG dataset (**train and test on one subject**). The test set contains 200 classes and performance is evaluated using Top-1 and Top-5 accuracies. We present a comprehensive comparison of different model types (EEG foundation models, time series models, and EEG-image models). NICE (EVA) denotes the integration of the NICE model into our proposed EVA framework for testing. The best result is highlighted in **bold**.

| Method | Neuro-GPT (Finetune) | CBraMod (Finetune) | Dlinear | NICE | ATM-S | NICE (EVA) | EVA (Ours) |
|---|---|---|---|---|---|---|---|
| Top-1 retrieval accuracy (0.5% chance level) | | | | | | | |
| Subject 1 | 0.3 | 24.5 | 24.0 | 16.0 | 16.5 | 19.7 | **31.0** |
| Subject 2 | 0.8 | 16.5 | **26.0** | 16.2 | 18.5 | 17.2 | 25.5 |
| Subject 3 | 5.0 | 23.5 | 25.5 | 20.8 | 21.5 | 25.3 | **33.0** |
| Subject 4 | 9.0 | 19.5 | 26.5 | 26.8 | 22.0 | 28.5 | **36.5** |
| Subject 5 | 0.5 | 9.5 | 12.0 | 12.7 | 16.5 | 15.8 | **21.5** |
| Subject 6 | 5.0 | 22.5 | 21.5 | 20.0 | 20.5 | 21.8 | **27.0** |
| Subject 7 | 16.0 | 13.0 | **29.0** | 21.0 | 22.0 | 22.2 | 28.5 |
| Subject 8 | 19.5 | 25.0 | 34.5 | 25.7 | 33.5 | 35.3 | **44.0** |
| Subject 9 | 11.0 | 16.5 | 10.0 | 19.0 | **27.0** | 16.8 | 26.5 |
| Subject 10 | 15.5 | 26.0 | 23.5 | 22.7 | 29.0 | 25.8 | **32.0** |
| Average | 8.3 | 19.7 | 23.3 | 20.1 | 22.7 | 22.8 | **30.6** |
| Top-5 retrieval accuracy (2.5% chance level) | | | | | | | |
| Subject 1 | 0.9 | 48.5 | 54.0 | 41.3 | 44.5 | 50.5 | **61.5** |
| Subject 2 | 30.0 | 44.0 | 54.0 | 47.8 | 44.0 | 45.0 | **59.0** |
| Subject 3 | 14.0 | 50.5 | 63.0 | 48.2 | 48.5 | 58.5 | **67.0** |
| Subject 4 | 25.0 | 45.5 | **61.0** | 59.8 | 52.0 | 60.3 | 60.5 |
| Subject 5 | 2.5 | 25.5 | 40.5 | 33.3 | 45.0 | 36.5 | **46.0** |
| Subject 6 | 19.0 | 49.0 | **55.5** | 51.3 | 52.0 | 52.8 | 52.5 |
| Subject 7 | 39.0 | 38.0 | 61.0 | 54.5 | 56.5 | 48.3 | **62.5** |
| Subject 8 | 46.5 | 54.5 | 70.5 | 60.5 | 67.0 | 70.5 | **72.5** |
| Subject 9 | 26.5 | 39.5 | 29.5 | 45.3 | **54.5** | 43.3 | 52.5 |
| Subject 10 | 51.5 | 58.0 | 58.0 | 52.2 | 65.0 | 63.2 | **65.0** |
| Average | 25.5 | 45.3 | 54.7 | 49.4 | 52.9 | 52.9 | **59.9** |

## A.4 ADDITIONAL RESULTS FOR IMAGE RETRIEVAL AND CLASSIFICATION

Table 3 presents the evaluation results for zero-shot retrieval tasks on the THINGS-EEG dataset, comparing our EVA framework against several existing approaches, including EEG foundation models (Neuro-GPT, CBraMod), time series models (Dlinear), and EEG-image models (NICE, ATM-S),

based on Top-1 and Top-5 accuracy across 10 subjects for 200 distinct classes. Our proposed EVA demonstrates a significant improvement over all compared methods, achieving the highest average Top-1 accuracy of 30.6% and an average Top-5 accuracy of 59.9%. This markedly surpasses the next best performing models, such as Dlinear (23.3% Top-1, 54.7% Top-5) and ATM-S (22.7% Top-1, 52.9% Top-5). The table also shows that integrating the NICE model within our EVA framework (NICE (EVA)) yields improved performance over standalone NICE, though our end-to-end EVA solution provides the most substantial gains, underscoring its superior capability in aligning EEG signals with visual semantic content in a challenging zero-shot scenario.

Table 4 presents a comprehensive evaluation of the zero-shot image retrieval task on the THINGS-EEG dataset using a leave-one-subject-out (LOSO) cross-validation approach, with performance assessed across Top-1, Top-5, 2-way, 4-way, and 10-way accuracies against their respective chance levels (0.5%, 2.5%, 50.00%, 25.00%, and 10.00%). Our proposed EVA demonstrates strong results, achieving the highest Top-1 accuracy of 12.40%. While ATM-S leads in Top-5 (33.73%) accuracy, EVA remains highly competitive with 30.25% in this category. These findings underscore EVA's robust generalization for image retrieval from EEG features of unseen subjects, outperforming various EEG-specific models like EEGNetV4 (6.25% Top-1) and EEG foundation models such as CBraMod (Finetune) (6.60% Top-1), and also showing a clear advantage over the NICE model even when integrated within our framework (NICE (Our Framework), 8.70% Top-1).

Table 4: Overall performance (accuracy %) of zero-shot image retrieval task based on THINGS-EEG dataset (**leave one subject for test**). The test set contains 200 classes and performance is evaluated using Top-1 and Top-5 accuracies as well as 2-way, 4-way and 10-way accuracies. The best result is highlighted in **bold**.

| Model Type | Methods | Top-1 | Top-5 | 2-Way | 4-Way | 10-Way |
|---|---|---|---|---|---|---|
| | Chance level | 0.50 | 2.50 | 50.00 | 25.00 | 10.00 |
| EEG Specific Model | MLP | 4.46 | 15.26 | 75.80 | 55.08 | 34.05 |
| | EEGNetV4 | 6.25 | 20.95 | 82.85 | 64.65 | 42.35 |
| | EEG Conformer | 0.87 | 4.42 | 56.54 | 31.80 | 13.89 |
| | ShallowFBCSPNet | 2.51 | 12.03 | 75.76 | 53.63 | 31.43 |
| EEG Foundation Model | CBraMod (Finetune) | 6.60 | 20.30 | 80.25 | 61.45 | 42.55 |
| | FoME (Finetune) | 3.57 | 10.43 | 62.50 | 48.51 | 29.35 |
| EEG-Image Model | NICE | 6.20 | 21.40 | –.– | –.– | –.– |
| | NICE (Our Framework) | 8.70 | 26.10 | 84.50 | 67.35 | 49.10 |
| | ATM-E | 7.00 | 21.12 | 80.65 | 61.65 | 39.66 |
| | ATM-S | 11.84 | **33.73** | 87.36 | **72.80** | 53.80 |
| | EVA (Ours) | **12.40** | 30.25 | **88.50** | 72.50 | **59.13** |

Further extending the evaluation, Table 5 details the performance on the zero-shot text retrieval task from THINGS-EEG data, where models were trained and tested on individual subjects. In this distinct task, EVA exhibits superior performance across all metrics, securing the top results with 10.85% Top-1 accuracy, 28.05% Top-5 accuracy, 84.70% 2-way accuracy, 69.55% 4-way accuracy, and 49.25% 10-way accuracy. This consistent lead highlights EVA's strong capability in aligning EEG signals with textual semantic representations. Compared to other models, including ATM-S (7.55% Top-1, 22.60% Top-5) and NICE (Our Framework) (7.25% Top-1, 26.60% Top-5), EVA again demonstrates a clear advantage, reinforcing its effectiveness in diverse zero-shot retrieval scenarios from EEG.

## A.5 ADDITIONAL RESULTS FOR VIDEO CLASSIFICATOIN

Metric Definitions in Table 1:

- "F / S" (Optical Flow Score): The optical flow score (OFS) of each 24 FPS two-second video clip obtained by averaging the length of the optical flow vectors, ranging from 0.008 (almost static) to 6.252 (rapidly changing). Further, based on the OFS, we divide all the

Table 5: Overall performance (accuracy %) of zero-shot text retrieval task based on THINGS-EEG dataset (**train and test on one subject**). The best result is highlighted in **bold**.

| Model Type | Methods | Top-1 | Top-5 | 2-Way | 4-Way | 10-Way |
|---|---|---|---|---|---|---|
| | Retrieving text using EEG features (train and test on one subject) | | | | | |
| | Chance level | 0.50 | 2.50 | 50.00 | 25.00 | 10.00 |
| EEG-Specific Model | MLP | 2.80 | 9.70 | 69.30 | 47.70 | 25.40 |
| | EEGNetV4 | 3.10 | 13.70 | 75.50 | 53.75 | 32.05 |
| EEG Foundation Model | BrainBERT (Probe) | 0.60 | 2.70 | 49.30 | 24.90 | 9.90 |
| | BrainBERT (Finetune) | 1.00 | 3.40 | 53.50 | 25.60 | 10.70 |
| | Neuro-GPT (Probe) | 0.45 | 2.75 | 69.30 | 42.60 | 23.50 |
| | Neuro-GPT (Finetune) | 1.72 | 9.25 | 49.80 | 24.50 | 9.60 |
| | CBraMod (Finetune) | 5.95 | 16.35 | 74.95 | 55.20 | 33.80 |
| EEG-Image Model | NICE (Our Framework) | 7.25 | 26.60 | 83.40 | 66.85 | 46.65 |
| | ATM-S | 7.55 | 22.60 | 82.75 | 65.40 | 43.25 |
| | EVA (Ours) | **10.85** | **28.05** | **84.70** | **69.55** | **49.25** |

video clips into 2 categories: Fast, Slow. We choose the median OFS of 1.799 as the threshold to make sure the label is balanced.

- "N. Obj" (Object Number): The number of the main objects. There are 3 categories: One, Two, Many. Many indicates the number of the main objects is equal to or more than three.

Table 6: Ablation study on the SEED-DV dataset: impact of sliding window parameters (Size, Stride) and fusion coefficient on **Top-1** classification accuracy (%). Results are shown for individual subjects (N=10) and their average.

| Method Subject | Sliding Window (Size, Stride) & Fusion Coefficient | | | | | |
|---|---|---|---|---|---|---|
| | (50, 25) & 0.1 | (100, 30) & 0.1 | (125, 100) & 0.1 | (100, 30) & 0.5 | (100, 30) & 1.0 | (100, 30) & 1.5 |
| Subject 1 | 10.52 | 12.19 | 8.33 | 12.60 | **14.06** | 13.65 |
| Subject 2 | 6.15 | 7.50 | 8.44 | 7.92 | **9.48** | 8.85 |
| Subject 3 | 5.73 | 6.88 | **7.08** | 5.42 | 6.25 | 5.63 |
| Subject 4 | 4.48 | 4.58 | 5.42 | 4.38 | **5.42** | 5.42 |
| Subject 5 | 7.29 | 7.60 | 7.19 | 8.33 | **10.94** | 6.46 |
| Subject 6 | 5.94 | 5.73 | 6.35 | 6.25 | **6.67** | 5.94 |
| Subject 7 | 5.63 | 5.10 | 6.35 | 5.10 | 4.69 | **6.46** |
| Subject 8 | 7.71 | **9.58** | 8.96 | 8.44 | 6.88 | 8.13 |
| Subject 9 | 7.60 | 5.52 | 5.21 | 6.15 | **7.60** | 5.83 |
| Subject 10 | 5.73 | **6.98** | 6.25 | 6.88 | 6.77 | 6.46 |
| Average | 6.68 | 7.17 | 6.96 | 7.15 | **7.88** | 7.28 |

Tables 6, 7, and 8 detail extensive ablation studies conducted on the SEED-DV dataset to validate our parameter choices and component contributions. Specifically, Tables 6 and 7 assess the impact of varying sliding window parameters (Size, Stride) and the fusion coefficient on Top-1 and Top-5 classification accuracy, respectively. The results demonstrate that a sliding window of (100, 30) combined with a fusion coefficient of 1.0 achieves the highest average performance, yielding 7.88% Top-1 accuracy and 24.01% Top-5 accuracy. Furthermore, Table 8 evaluates the significance of individual model components (Frequency Enhancement, Attention Block, TCN). This component-wise ablation confirms that the full EVA model consistently outperforms variants lacking any of these key modules, with average Top-1 and Top-5 accuracies of 7.88% and 24.01% respectively, highlighting the integral role each component plays in the framework's overall efficacy. For instance, removing the Attention Block or TCN notably degrades performance, underscoring their critical contributions.

Table 7: Ablation study on the SEED-DV dataset: impact of sliding window parameters (Size, Stride) and fusion coefficient on **Top-5** classification accuracy (%). Results are shown for individual subjects (N=10) and their average.

| Method / Subject | Sliding Window (Size, Stride) & Fusion Coefficient | | | | | |
|---|---|---|---|---|---|---|
| | (50, 25) & 0.1 | (100, 30) & 0.1 | (125, 100) & 0.1 | (100, 30) & 0.5 | (100, 30) & 1.0 | (100, 30) & 1.5 |
| Subject 1 | 26.56 | 34.69 | 25.94 | 34.58 | **38.85** | 34.48 |
| Subject 2 | 21.56 | 23.65 | 21.56 | 23.85 | **24.58** | 22.40 |
| Subject 3 | 17.29 | 19.69 | **20.83** | 16.98 | 18.23 | 17.92 |
| Subject 4 | 18.02 | 15.63 | 20.42 | **20.52** | 18.13 | 17.40 |
| Subject 5 | 23.33 | 25.94 | 22.50 | 28.13 | **31.04** | 24.38 |
| Subject 6 | 19.06 | 20.10 | 20.31 | 20.21 | 19.79 | **20.42** |
| Subject 7 | 20.83 | 17.92 | 19.38 | 17.40 | 19.06 | **22.50** |
| Subject 8 | 22.81 | **28.65** | 24.06 | 26.46 | 25.21 | 25.00 |
| Subject 9 | **22.50** | 18.54 | 21.56 | 18.44 | 21.25 | 20.73 |
| Subject 10 | 22.08 | 20.63 | 20.10 | 21.15 | **23.96** | 22.29 |
| Average | 21.40 | 22.54 | 21.67 | 22.77 | **24.01** | 22.75 |

Table 8: Ablation study on the SEED-DV dataset: impact of model components on Top-1 and Top-5 classification accuracy (%). Results are shown for individual subjects (N=10) and their average.

| Method / Subject | w/o Frequency Enhancement | | w/o Attention Block | | w/o TCN | | EVA (Full) | |
|---|---|---|---|---|---|---|---|---|
| | Top-1 | Top-5 | Top-1 | Top-5 | Top-1 | Top-5 | Top-1 | Top-5 |
| Subject 1 | 15.00 | 37.08 | 11.87 | 29.27 | **16.04** | 37.08 | 14.06 | **38.85** |
| Subject 2 | **9.58** | **25.52** | 7.92 | 19.69 | 6.67 | 23.54 | 9.48 | 24.58 |
| Subject 3 | 5.73 | 17.40 | 5.31 | 17.29 | **6.35** | 16.98 | 6.25 | **18.23** |
| Subject 4 | 4.38 | 15.63 | 3.96 | 17.60 | 5.31 | 15.10 | **5.42** | **18.13** |
| Subject 5 | 9.48 | 28.44 | 7.50 | 26.46 | 9.58 | 27.19 | **10.94** | **31.04** |
| Subject 6 | 7.29 | **23.02** | 6.25 | 19.37 | **7.40** | 18.13 | 6.67 | 19.79 |
| Subject 7 | 4.79 | 17.71 | 4.17 | 16.46 | **6.04** | 17.71 | 4.69 | **19.06** |
| Subject 8 | 6.67 | 21.88 | **8.33** | 25.00 | 5.10 | 18.85 | 6.88 | **25.21** |
| Subject 9 | 7.29 | 19.27 | 6.25 | 16.77 | 4.38 | 15.10 | **7.60** | **21.25** |
| Subject 10 | 6.15 | 18.96 | **8.33** | **24.06** | 6.46 | 18.65 | 6.77 | 23.96 |
| Average | 7.64 | 22.49 | 6.99 | 21.20 | 7.33 | 20.83 | **7.88** | **24.01** |

### A.6 MORE RECONSTRUCTION ANALYSIS AND CASES

We provide 40 image reconstruction results, as shown in Fig. 9, which presents comparisons between the reconstructed images, ground-truth stimuli, and the state-of-the-art method ATM-S. The features derived from EVA lead to reconstructions that more accurately capture the key semantic elements, shapes, and colors of both simple and challenging original stimuli. Compared to ATM-S, EVA-based reconstructions typically exhibit clearer object forms, more appropriate color palettes, and better overall similarity to the ground-truth images, whether the objects are common and simple, such as food or animals in straightforward scenes, or more complex, such as vehicles and detailed clothing in diverse backgrounds.

Table 9 offers a quantitative comparison of our EVA against other EEG-to-image methods, using a range of both low-level and high-level metrics. For low-level image fidelity, EVA demonstrates superior performance by achieving the highest pixelwise correlation (PixCorr) of 0.173 and the best structural similarity index (SSIM) of 0.372. In the high-level semantic comparisons, which primarily involve two-way identification accuracy (with a 50% chance level) using features from AlexNet (layers 2 and 5), Inception, and CLIP, alongside the SwAV average correlation distance (where

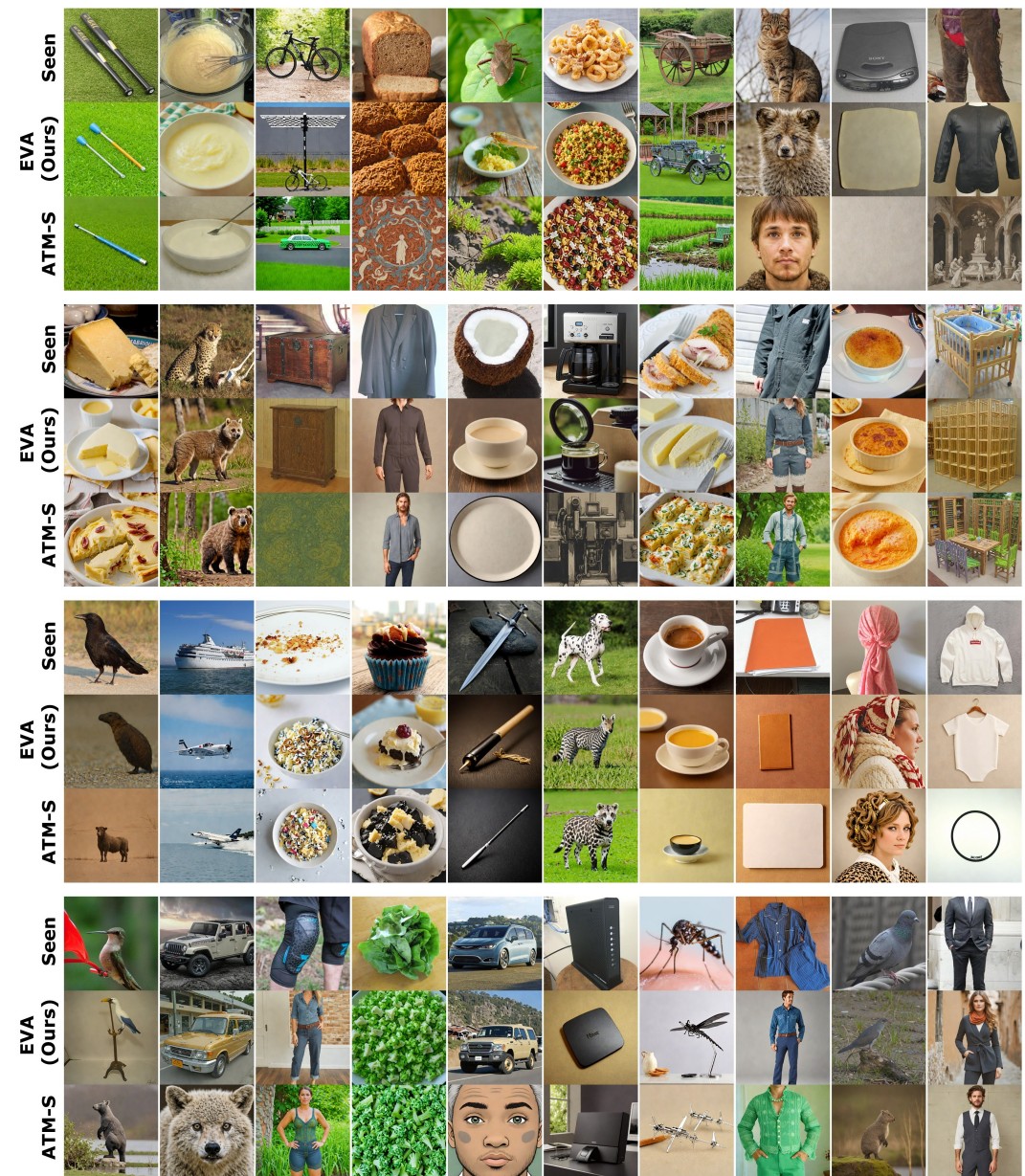

Figure 9: Additional results of zero-shot image reconstruction from the THINGS-EEG dataset. For more reconstructed original images, please refer to the supplementary materials.

lower is better), EVA again leads in most categories. Specifically, our model achieves top scores for AlexNet(2) (0.788), AlexNet(5) (0.878), CLIP (0.791), and the lowest (best) SwAV distance (0.578). While ATM shows a marginally higher score for Inception-based identification (0.734 vs. EVA's 0.730), our EVA framework consistently outperforms the other listed methods across the majority of metrics, indicating its enhanced capability in accurately reconstructing both the structural details and semantic content of images from EEG signals.

## A.7 ADDITIONAL RESULTS FROM ABLATION STUDIES

Fig. 10 demonstrates the performance comparison of different bandpass filter parameters in the Frequency-aware Dynamic Encoding (FADE) method across various frequency ranges. The nested bar chart illustrates both Top-1 and Top-5 retrieval classification accuracies, where the darker purple

Table 9: Quantitative comparison of EEG-to-image methods. PixCorr denotes the pixelwise correlation between ground truth and reconstructions; SSIM represents the structural similarity index metric; SwAV indicates the average correlation distance. All other metrics refer to two-way identification (with a 50% chance level). Two-way identification measures the percentage of correct decisions when comparing whether the original image embedding is more similar to its corresponding EEG embedding or to a randomly selected EEG embedding.

| Methods | Low-level | | High-level | | | | |
|---|---|---|---|---|---|---|---|
| | PixCorr↑ | SSIM↑ | AlexNet(2)↑ | AlexNet(5)↑ | Inception↑ | CLIP↑ | SwAV↓ |
| NICE | 0.142 | 0.276 | 0.739 | 0.832 | 0.659 | 0.722 | 0.612 |
| EEGNetV4 | 0.140 | 0.302 | 0.767 | 0.840 | 0.713 | 0.773 | 0.581 |
| ATM | 0.160 | 0.345 | 0.776 | 0.866 | **0.734** | 0.786 | 0.582 |
| EVA (Ours) | **0.173** | **0.372** | **0.788** | **0.878** | 0.730 | **0.791** | **0.578** |

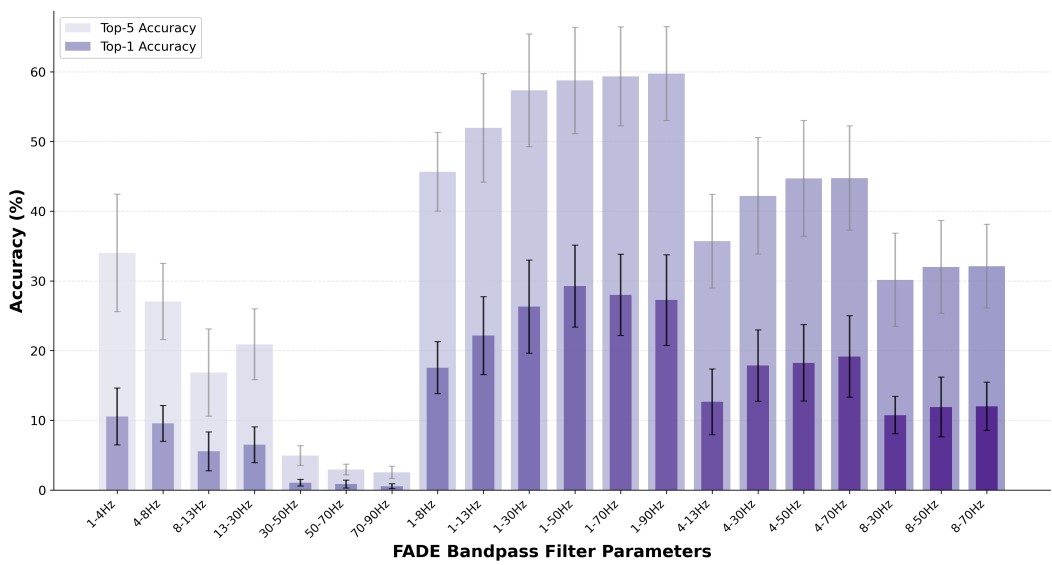

Figure 10: Performance comparison of FADE bandpass filter parameters showing Top-1 (dark purple) and Top-5 (light purple) retrieval classification accuracies with standard deviation error bars.

bars represent Top-1 accuracy nested within the lighter purple bars showing Top-5 accuracy. The results reveal that broader frequency ranges generally yield superior performance, with the 1-90Hz configuration achieving the highest Top-5 accuracy of 59.75% (±6.74%) and the 1-50Hz range delivering the best Top-1 accuracy of 29.25% (±5.87%). Notably, low-frequency components (1-4Hz, 4-8Hz) demonstrate substantial contribution to classification performance, while high-frequency ranges (50-70Hz, 70-90Hz) show limited effectiveness with Top-1 accuracies below 1%. The comprehensive frequency range of 1-70Hz and 1-90Hz configurations exhibit comparable performance, suggesting that frequencies above 70Hz provide minimal additional discriminative information. These findings indicate that FADE's effectiveness is primarily driven by low and mid-frequency neural oscillations, with optimal performance achieved when incorporating the full spectrum from 1Hz to approximately 50-90Hz.

## A.8 RATIONALE FOR THE SEPARATE CLASSIFICATION MODULE

In an ideal scenario, a robust alignment module should be sufficient to enable zero-shot classification without a separate head. As shown in Table 5, using only the alignment module with a text-based prompt, EVA achieves the best zero-shot classification performance among all compared methods.

However, we find that in practice, the feature space learned through contrastive alignment with visual features may not be perfectly optimized for supervised classification tasks with fixed, discrete label sets. The separate classification module is therefore a standard and effective practice in representation learning. It functions as a task-specific "head" that fine-tunes the general-purpose features from our encoder, ensuring optimal performance on specific benchmarks. This design allows our core EEG encoder to remain versatile while accommodating the specific requirements of different downstream tasks.

## A.9 USE OF LARGE LANGUAGE MODELS

Large Language Models (LLMs) were used in a limited capacity during the preparation of this manuscript. Specifically, LLMs were employed solely for language polishing and text refinement purposes, including:

- Grammar checking and correction of minor linguistic errors
- Improving sentence structure and clarity for better readability
- Ensuring consistent terminology usage throughout the manuscript

LLMs were **not** involved in:

- Research ideation or conceptual development
- Experimental design or methodology formulation
- Data analysis or interpretation of results
- Generation of technical content or scientific claims
- Writing of core technical sections or novel contributions

All scientific ideas, methodological innovations, experimental results, and technical contributions presented in this work are entirely the product of the authors' original research. The authors take full responsibility for all content in this manuscript, including any text that was refined using LLMs. The use of LLMs was limited to improving the presentation and clarity of the authors' original ideas and findings.

