# OpenReview forum: "EVA: Achieving Discriminative and Semantically Faithful Multi-Scale EEG-Vision Alignment"
_ICLR.cc/2026/Conference — ICLR 2026 Conference Withdrawn Submission_

### Official Review · Reviewer_oPcv · 2025-10-29

**Soundness:** 2
**Presentation:** 2
**Contribution:** 2
**Rating:** 4
**Confidence:** 2

**Summary:**

The paper presents a commendable and ambitious effort to create a unified framework for EEG-vision alignment across multiple modalities (images, videos, 3D objects). The core idea of balancing feature discriminability and semantic fidelity is well-motivated and addresses a significant gap in the field. However, while the empirical results are strong, the paper faces significant criticism regarding the perceived novelty of its technical components and the depth of its evaluation.

**Strengths:**

Well articulated problem statement
The experiments are strong

**Weaknesses:**

Pls see questions

**Questions:**

A major point of contention is the fundamental novelty of the core modules. More justifications on technical contributions should be provided.

The criticism that converting EEG to the frequency domain via rFFT is not novel is valid. While the introduction of an adjustable, learnable band-pass filter is a refinement, the core concept of frequency-domain analysis is a well-established technique in EEG processing, both in traditional signal processing and deep learning.

Although the specific integration of learnable clusters with cross-attention and a differentiable loss is sophisticated, the broader idea of dynamically grouping channels has been explored. More details would be good to provide, e.g., how ACC provides a qualitative leap over prior channel-dependent, independent, or clustering-based methods, rather than being an incremental improvement.

The paper identifies the low SNR of EEG as a key challenge but does not sufficiently explain how EVA's modules specifically and effectively solve this long-standing problem in this field. FADE's filtering is a known strategy, and it's unclear if the adaptive aspect provides a decisive advantage over carefully designed fixed filters in combating noise. A more direct comparison or analysis of noise robustness is needed.

While the paper compares against a reasonable selection of models, the field of EEG decoding is vast. The absence of comparisons with other recent, powerful architectures or signal processing techniques leaves room for doubt about whether the performance gains are due to a genuinely superior approach or a less comprehensive benchmark.

The framework, particularly the ACC module, introduces significant complexity with multiple new hyperparameters, e.g., number of clusters, temperature τ, annealing parameter γ, loss coefficient λ. The paper would be strengthened by a discussion on the robustness of these parameters and the practical challenges of tuning such a system.

---

### Official Review · Reviewer_g226 · 2025-11-01

**Soundness:** 3
**Presentation:** 2
**Contribution:** 3
**Rating:** 4
**Confidence:** 3

**Summary:**

This paper addresses a critical gap in EEG-based visual decoding: the lack of unified frameworks capable of handling heterogeneous visual stimuli (images, videos, 3D objects), and balancing the features' discriminability and semantic fidelity. The proposed EVA fills this void via two core innovations: (1) the FADE module (frequency-aware dynamic encoding) extracts compact, noise-resistant frequency features via adjustable band-pass filtering and (2) the ACC module (adaptive channel clustering) dynamically groups EEG channels by functional synergy rather than fixed anatomy. Experimental results reveal the effectiveness of FADE in classification, retrieval, and reconstruction tasks.
Overall, this paper could be a nice contribution to EEG-based visual decoding, with the clarifications on the motivation and comprehensive experiments across datasets. I have some concerns about the presentation and analysis of this paper, but I would be willing to increase the score with the author's clarification.

**Strengths:**

- EVA unifies multi-scale EEG signal alignment with heterogeneous visual stimuli—images, videos, and 3D objects—within a single contrastive learning-based architecture, which makes it a universal solution for various tasks.
- The author proposes an ACC module to capture inter-channel synergies without relying on fixed anatomical groupings. Experimental analysis also reveals distinct processing strategies across clusters.
- The presentation of this paper should be improved: e.g., lack of analysis and explanation in Figures 7-8, line 290 is incomplete, tiny fonts in Figure 7, and Algorithm 1 should be moved to the main part.

**Weaknesses:**

- The motivation of improving discriminability and semantic fidelity at the same time is not new in other ML domains; the author should add more discussions on the typical challenges when applying it to EEG-based visual decoding.
- Lack of quantitative metrics for the image reconstruction task.
- The presentation of this paper should be improved: e.g., lack of analysis and explanation in Figures 7-8, line 290 is incomplete, tiny fonts in Figure 7, algorithm 1 should be moved to the main part, some of the method part is confusing.

**Questions:**

- I am confused about formula (3), how does the cluster update via cross attention, and what is the optimization target?
- How do you split the dataset? e.g., in the Things-EEG dataset, do the 200 candidates appear in the training data?
- What is the parameter ablation in Figure 7? What is the conclusion in your sensitive analysis?

---

### Official Review · Reviewer_6VKM · 2025-11-02

**Soundness:** 2
**Presentation:** 2
**Contribution:** 2
**Rating:** 4
**Confidence:** 4

**Summary:**

This paper introduces a framework for EEG-Vision Alignment (EVA), a comprehensive approach that aims to align EEG signals with various types of visual stimuli, including images, videos, and 3D objects. The primary challenge it addresses is the inability of current methods to generalize effectively across different modalities and temporal scales.

**Strengths:**

+ The framework is designed to unify alignment across various visual stimuli, including images, videos, and 3D objects, addressing a limitation present in many existing models.

+ The Universal EEG Encoder consists of two key components: FADE, which utilizes frequency-domain information adaptively, and ACC, which groups channels to manage noise. These components target fundamental challenges in EEG processing.

+ The framework optimizes for both discriminative (classification) and semantic (reconstruction) tasks, which may enhance the robustness and meaningfulness of EEG representations for various downstream applications.

**Weaknesses:**

- Referring to the encoder as "Universal" is a significant assertion. While this designation suggests progress in achieving generalizability, its actual effectiveness may still be limited by the specific types of data and tasks on which it was trained.

- The integration of FADE and ACC components within the encoder is likely to increase the model's computational complexity and the number of hyperparameters that require tuning. This could present additional challenges during the optimization process.

- Although the framework is presented as universal, it is plausible that the task-specific adaptations mentioned are contributing significantly to performance, indicating that the core encoder's universality may not be as robust as initially suggested.

**Questions:**

Please address the concerns in the areas of weakness mentioned above.

**Details Of Ethics Concerns:**

nil

---

### Official Review · Reviewer_JbsJ · 2025-11-03

**Soundness:** 2
**Presentation:** 3
**Contribution:** 2
**Rating:** 4
**Confidence:** 3

**Summary:**

This paper studies EEG-vision tasks, where the authors argue that the existing studies fail to investigate powerful EEG encoders and align EEG data with different visual modalities. Accordingly, the authors propose an EEG encoder, through which the multichannel EEG data elicited by heterogeneous visual stimuli (including 100 ms RSVP images, 2 s videos, and 1 s 3D object rotations) maps into a representation space that supports visual retrieval, classification, and reconstruction. Specifically, the EEG encoder applies rFFT/irFFT to capture frequency features and learns soft channel clusters using masked cross-attention. The overall training objectives combines an MSE fidelity term to pre‑trained vision(-language) embeddings, a contrastive discriminability term, and a structural regularizer over channels. The pipeline attaches a task‑specific classification head, an alignment module trained contrastively, and a reconstruction prior that regresses image features later used with IP-Adapter and SDXL-Turbo.

Empirically, the proposed method (named EVA) improves zero‑shot THINGS‑EEG image retrieval with per‑subject averages 30.6 top-1 / 59.9 top-5 and a cross-subject LOSO top-1 of 12.40. On SEED-DV video classification, EVA achieves higher accuracy than baselines. On EEG-3D, EVA slightly outperforms baselines. The authors also study semantic fidelity through multi-frame alignment and zero-shot image reconstruction.

**Strengths:**

1. The proposed method addresses the limitations of existing task-specific, time-domain focused, and static channel operation-based EEG encoders, by proposing a frequency-aware encoder with learnable channel‑clustering, then validates the discriminability–fidelity framing across three datasets.
    2. The empirical improvements on THINGS-EEG are consistent.
    3. The ablations are informative, which isolate FADE/ACC effects and key head components.

**Weaknesses:**

1. The improvements applied to EEG encoder, namely FADE that takes into account frequency domain information and ACC that makes channel clustering learnable, are not entirely new concepts in time-series data representation learning, even in the context of EEG.
2. To make the system be able to handle multiple tasks, the EVA framework is designed to optimize for multiple loss functions simultaneously. However, there is no analysis in terms of the relatedness of these objective functions, nor  treatments to handle the (potential) conflicts often happening in multi-task learning.
3. The $\mathcal{L}_{\rm struct}$ used by ACC was not analyzed for normalization assumption on the channel similarity matrix $\mathbf{S}$ or for avoidance of degenerate minima beyond an entropy term. I think a connection to normalized-cut objectives would help.
4. Statistical reporting is thin on the main results. The top-k accuracies are given without seed variability (i.e., multiple runs with different random seeds) and confidence intervals. In this case, several 1 - 2 percentage improvements (e.g., some EEG-3D metrics in Table 2) could plausibly be attributed to noise.
5. Cross-subject generalization is still limited, which has often been the primary concern in the EEG community. While the paper acknowledges personalization as future work, a small “adapter”-like transferability attempt would elevate the practical utility of the proposed method.

**Questions:**

To solidify the paper and potentially raise my score, I request the following clarifications:
1. How does the proposed EEG encoder compare against existing EEG studies that also take into account frequency domain feature and adaptive channel aggregation in terms of the difference in underlying principle and insights?
2. Have the authors observed task conflicts when the method was optimizing for all loss functions simultaneously?
3. For ACC, state the assumptions on the similarity matrix and provide a short derivation linking eq.12 to normalized-cut objectives, explaining how trivial solutions are avoided beyond the entropy term.
4. Provide a compute parity table for baselines, covering epochs, early stopping, hyper parameter grids, etc; and indicate which methods were retrained under your preprocessing vs. taken from prior reports directly.

---

### Note · Authors · 2025-11-28

I have read and agree with the venue's withdrawal policy on behalf of myself and my co-authors.